# Strengths and Weaknesses of the Pharmacovigilance Systems in Three Arab Countries: A Mixed-Methods Study Using the WHO Pharmacovigilance Indicators

**DOI:** 10.3390/ijerph19052518

**Published:** 2022-02-22

**Authors:** Hamza Garashi, Douglas Steinke, Ellen Schafheutle

**Affiliations:** Division of Pharmacy and Optometry, School of Health Sciences, Faculty of Biology Medicine and Health, The University of Manchester, Manchester M13 9PT, UK; douglas.steinke@manchester.ac.uk (D.S.); ellen.schafheutle@manchester.ac.uk (E.S.)

**Keywords:** pharmacovigilance, adverse drug reactions, Arab world, developing countries, program evaluation

## Abstract

Using the WHO pharmacovigilance (PV) indicators as a framework, this study aimed to explore the structures, processes, and outcomes of three Arab countries’ (Jordan, Oman, and Kuwait) PV systems to inform recommendations for countries with nascent PV systems. A mixed-methods design involving document review, semi-structured interviews, and a questionnaire was employed. Fifty-six key informants from the three countries’ national PV centres (NPVCs) and pharmaceutical industry were interviewed. The questionnaire collecting quantitative measures was only completed by Oman and Kuwait’s NPVCs. Using the framework, system strengths were attributed to the presence of “core” structural indicators, including a dedicated and officially recognised NPVC, PV legislation, and a national PV advisory committee, as well as “complementary” structural indicators, e.g., a computerised case-report management system. Contrastingly, weaknesses were attributed to the absence of these indicators plus other “core” structural indicators, namely, regular financial provision and adequate staff. Other weaknesses were attributed to low performance in “core” process and outcome indicators including reporting rates, reporter awareness, and signal detection. Greater governmental prioritisation through the provision of legislative enforcements, resources, and expertise as part of a well-structured system is required. More regional coordination efforts are needed to allow for sharing of expertise in order to bolster nascent systems.

## 1. Introduction

Pharmacovigilance (PV) is defined by the World Health Organisation (WHO) as “the science and activities relating to the detection, assessment, understanding and prevention of adverse effects or any other drug-related problem” [1] (p. 7) and represents an important element of a country’s public health policies’ portfolio. PV has four main objectives, namely, (a) the identification and quantification of previously unknown drug safety hazards; (b) the elucidation of predisposing factors to drug safety hazards, which if avoided could improve drug safety; (c) obtaining safety evidence on approved drugs to widen their usage; and (d) refuting false-positive adverse drug reaction (ADR) signals [2].

Increased efforts towards addressing public health concerns have led to an unprecedented expansion in global access to healthcare and medicines over the past decades [3]. However, these efforts have not been met by a proportionate improvement in PV systems in developing countries. A systematic review of developing countries’ PV systems found that overall system performance was poor and varied widely from one country to another [4]. This means that these systems are unable to benefit from local data identifying ADR signals to support regulatory decisions regarding drug safety in the populations they serve [3].

Like other developing countries, PV systems in Arab countries are at different stages of maturity, with many still in the early stages of development [5,6,7]. Recently, however, the importance of having a strong PV system in place has gained increased attention [8]. Recognising the importance of PV as a part of public health, the Arab League developed the “Guideline on Good Pharmacovigilance Practices (GVP) for Arab Countries” in 2014 with the aim of harmonising practices in the region [6,8,9]. However, effective implementation of these guidelines requires improvement in the existing PV systems in these countries. Literature on PV systems in the Arab World has mainly focused on surveying these countries’ systems and providing a descriptive overview of their characteristics [5,7,10]. However, no studies have been conducted that set out to provide an in-depth exploration of the PV situation within the individual countries.

An important function of policymakers involves maintaining oversight over implemented policies to ensure their efficiency and effectiveness [11]. Policy analysts and policymakers have long held an interest in cross-country comparisons of health systems and policies as understanding systems, processes, and developments in one group of countries can help inform policy learning and implementation in another [12].

To ensure patient safety and enhance efforts aimed at supporting the development and strengthening of PV systems in Arab countries, we see an imperative to understand existing conditions within the individual countries. Therefore, this study aimed to explore the structures, processes, and outcomes of three Arab countries’ PV systems, with differing levels of performance, to identify their areas of strength and weakness to inform recommendations, which will lead to the strengthening of their PV systems as well as those of other Arab and developing countries with nascent PV systems.

## 2. Materials and Methods

### 2.1. Study Design and Setting

This was a cross-sectional mixed method study employing qualitative and quantitative methods conducted in two parts in 2019 and 2021. The first part consisted of a documentary review and semi-structured interviews with key informants. The second part consisted of a questionnaire. The qualitative phase helped the researchers capture details of PV system implementation methods and gain insights into participants’ perceptions of their countries’ PV systems’ strengths and weaknesses, which helped in understanding the multifactorial issues affecting system performance. The quantitative phase involved collecting quantitative measures of PV system processes and outcomes and served to support the interview data in fully capturing the countries’ PV systems’ performance.

The study was set in three Arab countries, namely, Jordan, Oman, and Kuwait, whose surface areas are 89,320, 309,500, and 17,820 km^2^ [13], with total populations of 10,203,140, 5,106,622, and 4,270,563 persons, respectively [14].

### 2.2. Theoretical Framework

This study utilised the WHO PV indicators as a framework to gain insights into how PV systems are being implemented, as well as the factors impacting their performance in the three Arab countries. The WHO PV indicators “provide information on how well a PV programme is achieving its objectives” [15] (p. 4). They are designed to be simple enough to be understood without formal training in monitoring and evaluation [15]. There is a total of 63 indicators, which are divided into two main groups: core and complementary. Each of these types is further subdivided into three categories: structural, process, outcome/impact [15]. Further details can be found in the WHO PV indicators manual [15]. An alternative framework, the indicator-based PV assessment tool (IPAT), was considered but was dismissed due to its lack of sensitivity and specificity as a measurement tool [16].

### 2.3. Data Collection, Sampling, and Recruitment

Qualitative data were collected via the review of documents (e.g., annual reports, national PV regulations, and/or guidelines) which were either publicly available from the countries’ national medicines regulatory authority (NMRA) websites or provided by the study participants. In addition, semi-structured interviews were conducted with key informants from the national PV centres (NPVC) and the pharmaceutical industry (PI) in the three study countries. Purposive and snowball sampling was used for the semi-structured interviews. Potential participants were recruited via a gatekeeper at the respective NMRAs and asked to contact the researcher to participate. It was deemed sufficient to interview 12 to 20 individuals per country to achieve data saturation [17]. Audio recorded face-to-face interviews (in English) lasting 60 min on average were conducted by the primary author from April through December 2019. The interviews were conducted sequentially from highest (Jordan) to lowest (Kuwait) PV system performance. This sequential approach enabled the use of the insights gained from earlier in later interviews, particularly for sense checking potential recommendations in Kuwait. The interview guide (Appendix A) was informed primarily by the WHO PV indicator tool [15] and various other sources of literature surrounding PV with an enquiry of perceived system strengths and weaknesses. Participants were also asked to fill out a form providing background information about themselves (e.g., gender, education, employment sector).

Quantitative data were collected via purposive sampling after the conclusion of the qualitative data collection and analysis using a self-administered questionnaire (Appendix A) on PV system processes and outcomes (core and complementary) to gain a better understanding of the countries’ PV systems’ performance as per the WHO PV indicator tool [15]. The survey was sent via e-mail as a Microsoft Word document to the PV leadership in all three countries in July 2021, followed by monthly e-mail reminders until November 2021.

### 2.4. Data Analysis

Qualitative data obtained from the document review and verbatim interview transcripts were managed using NVivo 12 and subjected to thematic framework analysis involving five steps: familiarisation, coding, thematic framework identification, charting data into a matrix, and data interpretation [18]. The data were mapped onto themes informed by the WHO PV indicators [15].

Quantitative data obtained via the questionnaire were absolute numbers, percentages, and rates, which were entered into Microsoft Excel and calculated as determined by the relevant indicator. Scores were assigned to each category of indicators, which were then used to compare the countries on the basis of their total performance score. Each indicator was scored separately, and then a final score was calculated for each country on the basis of the 63 indicators. For the structural indicators, scores of 1, 0.5, or 0 were given, depending on whether the indicator satisfied, partially satisfied, or did not satisfy the WHO’s recommendations, respectively. For the process and outcome indicators, a score of 1 was given if the answer provided was >0; otherwise, it was scored as 0. Where an indicator (structure, process, or outcome) is made up of subset indicators, the score of 1 was divided equally among each of the subset indicators (e.g., where the indicator is divided into subset “a” and “b” each will be worth 0.5). The response data were tabulated and displayed as a radar chart to allow for visualisation of each country’s PV system’s performance.

### 2.5. Ethical Considerations

Approval to conduct the study was obtained from the University Ethics Committee, the three countries’ NMRAs, and members of the PI. All interview study participants were provided with a study information sheet to allow for an informed decision on whether to take part in the study, and they were asked to sign a written consent form. As the questionnaire participants were contacted by email, their completion and return of the questionnaire was taken as implied consent.

## 3. Results

Fifty-six participants were interviewed (17 in Jordan, 16 in Oman, and 23 in Kuwait). Only two interviews were not audio-recorded, wherein detailed notes were taken. All members (n = 5 per country) of the three countries’ NPVCs participated, as well as two members of the Jordanian regional PV centres. Most (n = 48) participants were pharmacists, and they were mainly (n = 38) employed by the PI. Questionnaires requesting data on the process and outcome indicators were only completed by the NPVCs in Oman and Kuwait (but not Jordan).

The three countries’ PV systems were evaluated for the 63 WHO PV indicators which contain 27 “core” and 36 “complementary” indicators. The PV systems of Jordan, Oman, and Kuwait achieved aggregate scores of 8, 11, and 11, respectively, for the “core” indicators and 9, 18, and 7, respectively, for the “complementary” indicators. The “process” and “outcome” indicators for Jordan which were not supplied were scored as 0. A complete breakdown of the total scores according to each group of indicators is provided as a visual representation in Figure 1. In what follows, the study findings for the three study countries are presented in two sections, namely, WHO “Core Indicators” and “Complementary Indicators” [15].

### 3.1. Core Indicators

#### 3.1.1. Core Structural Indicators

The WHO indicates that a prerequisite of a functional PV system is the presence of a dedicated space (i.e., a centre, department, or unit) for PV activity, which is officially recognised and/or accredited by the country’s Ministry of Health (MOH). Only in Kuwait was the NPVC not officially recognised by the country’s MOH and hence operated as an unofficial unit (sub-section) of the NMRA’s Drug Registration Department. Some NPVC and PI participants from Kuwait believed that their system’s lack of a dedicated and officially recognised NPVC represented a weakness because it resulted in a lack of authority and autonomy and prevented the system from being operational. In contrast, a few NPVC and PI participants in Jordan and Oman pointed to their countries’ PV centres’ official recognition as a strength since it provided it with increased visibility and significance. See Table 1 for a summary of the results for this group of indicators.


*“This [the establishment of an official PV department as a strength] is because it was a section of a department before, therefore was not that much importance placed on the section in terms of the reports received and increasing their numbers.”*
(Participant 1, NPVC, Oman)


*“The lack of a dedicated PV department is a weakness… the dedicated department is very important to act on a legal basis with proper staff, with proper infrastructure, with proper independent decisions, to have the full structure, full capacity to work with a proper PV system.”*
(Participant 17, NPVC, Kuwait)

A few members of Jordan’s NPVC believed that the NPVC’s affiliation with the NMRA was also a point of strength due to the NMRA’s extended reach and authority.


*“Being part of the regulatory body is good for PV in that you have the tools, you have the law, you can go see patient files, do further investigations within the hospitals. That’s why I think it’s our strength to be part of the regulatory body.”*
(Participant 7, NPVC, Jordan)

An important element of a PV system is the existence of an authoritative instrument, e.g., a national policy document or a legislative provision enacted by the appropriate arm of government to support PV activities. Only Jordan’s system possessed legislation for PV. A few participants from the Jordanian NPVC referred to the presence of a legal statute for PV as a strength of the system, which provided them with the necessary tools to monitor and enforce the implementation process across all stakeholders. A few participants from Kuwait’s NPVC and PI believed that the lack of a PV legal framework was a shortcoming as it meant that PV activities were undertaken without legal backing, thus preventing members of the NPVC from ensuring that pharmaceutical companies complied with decisions on PV, e.g., performing a leaflet change. Interestingly, none of Oman’s participants mentioned the absence of a statutory provision as a weakness as part of the discussion on system strengths and weaknesses.


*“…we feel tied up with the fact that we haven’t got a legal framework, so that’s a big weakness… the activities are being carried out, but the activities are being carried out with no umbrella, there’s nothing that protects them.”*
(Participant 11, NPVC, Kuwait)

A country’s medicines regulatory authority or agency acts as an important stakeholder and focal point for promoting PV. Jordan’s NMRA was the only one that was independent of the country’s MOH. A few participants from the PI in Jordan cited the NMRA’s autonomy in decision making and authority in dealing with pharmaceutical companies as a strength of the PV system.


*“The fact that the drug authority and the PV centre are separate from the MOH is, in my opinion, a strength. ... A drug authority, which is an entity that gives and takes back the marketing authorisation, are controlling the industry through this, so if you don’t report, and you don’t have a system, and you are not compliant with regulations, we have the authority to withdraw your marketing licence. The MOH does not have this authority.”*
(Participant 14, PI, Jordan)

Availability of a regular and sustained funding source is necessary for running a PV system. All three countries lacked a dedicated PV budget, and thus financial resources were obtained through the NMRA’s (Jordan) or the MOH’s (Oman and Kuwait) budgets. However, only a few participants from the NPVC, regional PV centres, and PI in Jordan commented on this issue as hindering activities such as training workshops for healthcare providers (HCPs) or awareness-raising campaigns. Hence, there was a reliance on obtaining funding from outside sources.


*“…we don’t have a budget for things like printing materials, conducting training outside. When you perform training outside you need coverage to sponsor the event, to provide meals for those attending. We don’t have a budget here at the Jordan Food and Drug Administration (JFDA) for our department for these activities. So, you need sponsors from outside to implement these things.”*
(Participant 2, NPVC, Jordan)

A PV system needs trained staff according to the expected total full-time equivalents required to enable the PV centre to fulfil all its essential duties and responsibilities. All three countries’ PV systems were similar in terms of the number of staff working at the NPVC. The three countries’ NPVC members and a few industry participants, as well as a few participants from Jordan’s regional PV centres, agreed that staff shortages were a weakness. This caused delays in work that must be done regularly or on a scheduled basis, e.g., entering ADR reports in the national database; review of PV reports, i.e., periodic safety update report (PSUR) and risk management plan (RMP); or publication of a bulletin/newsletter for PV information dissemination.


*“It’s [the lack of staff] affecting our work in that we have many PV activities to do, for example, we have to enter reports onto the VigiFlow, which should be done regularly, but is not. So, once we have time then we are entering our reports into VigiFlow. So, this is affecting our implementation, for example, we should by now have completed the inspection on all companies and all drug stores, but we have not. There is also training and awareness campaigns, which is not being done according to the scheduled program.”*
(Participant 2, NPVC, Jordan)


*“This [staff shortage] is the major factor, because for example when you want to study a PSUR you need teamwork to be able to do this quickly. The files for the PSUR are large. One person cannot review every file for every medicine. Also, we are receiving PSURs every six months for every medicine.”*
(Participant 1, NPVC, Oman)

A few of Jordan’s and Oman’s NPVC and industry participants pointed to the scarcity of individuals with PV expertise and staff turnover due to the large workload that came with working in PV which exacerbated this problem. This, in turn, meant a loss of continuity in terms of the team members working in the department in addition to the loss of time and effort spent in training them.


*“…the turnover of staff between the departments also, it is a weakness that we spend time and money to do training for [a] certain individual and then he will go to another department.”*
(Participant 7, NPVC, Jordan)

Oman was the only country where PV was incorporated into the national curriculum of HCPs (pharmacists and nurses), however none of the country’s participants commented on this issue as part of the discussion on system strengths and weaknesses. Contrastingly, a few industry participants from Jordan and Kuwait believed that PV’s lack of incorporation into HCPs curriculums was contributing to a lack of knowledge and awareness regarding ADR reporting among health workers and as such was a shortcoming of their respective countries’ PV systems.


*“…in other countries, HCPs’ awareness is very high. It is part of their education in the universities. Here, it’s not implemented yet, so the HCPs, they are shaky, shall we inform or not? How to report? When to report? What to report? Still, their awareness and the level of education… [has] not reached the level of other people [in other countries], so it’s still not high. The awareness level is not high.”*
(Participant 13, PI, Kuwait)

The WHO PV indicators include the existence of a qualified committee that can provide advice and technical assistance as an important component of the PV system. Only Jordan had a PV advisory committee consisting of HCPs representing different sectors. A few members of Jordan’s NPVC and regional PV centres viewed the presence of this committee, which provided advice to the NPVC on the basis of its members’ varied expertise as a strength. In comparison, a few members of Oman’s NPVC viewed the absence of such an advisory committee from their system as a weakness.


*“Another positive is the presence of the Health Hazard Committee, which has benefitted us a lot since it is composed of individuals representing different sectors and from different healthcare professions.”*
(Participant 6, NPVC, Jordan)


*“…I always think that we [the NPVC] are sitting in a remote position and we are not in the practising side… we are not able to find out whether it is the prejudice among the healthcare professionals or the patients that they say it is ineffectiveness, or whether it is actual ineffectiveness which is happening.”*
(Participant 5, NPVC, Oman)

#### 3.1.2. Core Process Indicators

The WHO identifies the number of ADR reports received annually as one of the measures of the PV system’s activity. The volume of reports generated within the population in Oman was higher than in Kuwait (31.88 versus 16.58 reports/100,000 population, respectively). Similarly, the WHO’s guidance refers to the number of cumulative reports in the national ADR database since its inception as another measure of system activity. Oman’s NPVC had collected more ADR reports since the PV systems’ inception than that of Kuwait (19,731 versus 890 reports, respectively). A few NPVC and regional centre participants in Jordan and the NPVC in Kuwait, in addition to some industry participants from Jordan, all cited low ADR-reporting rates as a weakness in their PV systems. Interestingly, a few industry participants from Oman mentioned that this was a problem mainly within the private healthcare sector. Low ADR reporting rates prevented the NPVCs from obtaining a clear view of ADR prevalence in the country and hindered making locally relevant drug safety decisions. Participants cited multiple reasons for the low ADR reporting, including poor knowledge, awareness, and/or attitude of reporters towards PV, as well as lack of mandatory HCP reporting legislation. See Table 2 for a summary of the results for this group of indicators.


*“Although HCPs may encounter patients with ADRs, some of them don’t know that [they have encountered an ADR], or some of them don’t know that they have to report it, or that it’s important to report it. So, I think that one weakness is that not all HCPs report ADRs.”*
(Participant 3, peripheral PV centre, Jordan)

Oman had a higher percentage of satisfactorily completed ADR reports submitted to their NPVC compared to Kuwait (84.3% versus 58.9% respectively), and unlike in Kuwait, these reports were submitted to WHO’s VigiBase. When asked about their views on the strengths and weaknesses of the PV system, a few members of Jordan’s regional PV centres and Kuwait’s NPVC cited poor quality of ADR reports as a weakness and thus a high percentage of the ADR reports received were of little value.


*“Even though we have 1000 reports, I believe that 70–80% of them are of poor quality. And personally, I know that in one year I provided the PV centre with more than 160 reports, and I later found out that only 40 of them were very useful. …But unfortunately, we never worked on the reports in terms of their quality, we never did statistics on the reports, we don’t know what the gap is, what is the problem with our reports, why are our reports not of good quality.”*
(Participant 4, regional PV centre, Jordan)

#### 3.1.3. Core Outcome Indicators

According to the WHO, the PV system’s ability to detect signals indicates its capability of ensuring drug safety. Neither Oman’s nor Kuwait’s NPVC had detected any signals during the past five years (Table 3). Interestingly, none of the participants in these two countries cited this issue as being a weakness of their respective countries’ PV systems. However, a few participants from Jordan’s NPVC, regional PV centres, and PI pointed out that the inability to detect signals arising from the data obtained through local ADR reporting was a weakness of the system. The lack of signal detection hampered drug safety decision making and was attributed to the low quantity and quality of submitted ADR reports.


*“One of the reasons [for the deficiency in signal detection] is that we don’t have enough data, quality data, and the people at the PV centre they focus on collecting the reports without taking it for a further step of analysis and investigation. I think this as well is an issue that our industry has because it is not only the duty of the healthcare system or the health authorities, but also one of the responsibilities of the MAH.”*
(Participant 4, regional PV centre, Jordan)

The WHO points out that the number of regulatory actions taken by the NPVC provides a measure of regulatory decisions made, on the basis of PV activities, to ensure drug safety. Although regulatory actions exclusively on the basis of local PV data were not taken by the NPVCs in both Oman and Kuwait, it was acknowledged that regulatory actions on the basis of a combination of local and other countries’ data had been taken. A few participants from Kuwait’s NPVC brought up this issue when discussing low ADR reporting rates as a weakness of their PV system.


*“…we need more reporting to have our own decision-making process based on our own data in Kuwait. We don’t want to depend on international data. We need to depend on our own data to take into consideration our lifestyle, our raised diet, concurrent medications, morbidity and so on, so that’s why this [i.e., under-reporting] is one of the weaknesses and one of the barriers that we need to overcome.”*
(Participant 17, NPVC, Kuwait)

### 3.2. Complementary Indicators

#### 3.2.1. Complementary Structural Indicators

Interviewees in Kuwait indicated that the NPVC did not have access to a computerised case-report management system, which hindered their ability to adequately analyse local data. On the other hand, a few NPVC interviewees from Jordan and Oman cited their centres’ use of the WHO-provided case-report management system VigiFlow as a strength because it provided them with a database for report management and storage as well as performing statistical analysis. See Table 4 for a summary of the results for this group of indicators.


*“…the IT system [is a weakness], it’s very important for our work to get a proper database and to have a system such as the VigiFlow or the VigiLyze and VigiBase to help get a broader vision of the different cases worldwide. For signal detection, it’s very important to have a system as well, to help get the proper signal as quickly as possible and as efficiently as possible.”*
(Participant 17, NPVC, Kuwait)

#### 3.2.2. Complementary Process Indicators

Neither Oman nor Kuwait possessed data on HCPs’ and patients’ awareness levels. Oman’s NPVC had organised more PV training sessions for HCPs and therefore trained more individuals compared that of Kuwait. However, neither country’s NPVC had organised training sessions for the public. A few participants from Oman’s NPVC and PI believed that a strength of their country’s PV system was the increased levels of awareness, particularly among HCPs, which was contributing to improved ADR reporting. This observation was attributed, in part, to the NPVC’s continuous efforts to increase awareness levels. In contrast, some industry participants from the three countries, a few NPVC participants from Jordan and Kuwait, and a few participants from Jordan’s regional PV centres mentioned a lack of awareness regarding PV among both HCPs and the public as a weakness. Participants further believed this to be one of the main reasons for the low ADR reporting rate. See Table 5 for a summary of the results for this group of indicators.


*“A point of strength is that there is now awareness. I feel the first step that we took was to increase awareness of HCPs and the general public. This resulted in us receiving many reports.”*
(Participant 10, NPVC, Oman)


*“...the awareness campaigns are still not strong enough. We don’t hear in Kuwait, I didn’t hear that there is a committee for PV or an awareness campaign, to increase awareness of the patients.”*
(Participant 13, PI, Kuwait)

#### 3.2.3. Complementary Outcome Indicators

Only in Oman were figures available on the percentage of preventable ADRs and medicine-related birth malformations, and both were low. No information was reported on the remaining eight indicators in this group for either Oman or Kuwait (Table 6). Figure 2 provides a visual illustration of the areas of strength and weakness of the three studied countries’ PV systems.

## 4. Discussion

To the best of the authors’ knowledge, this is the first study to employ the core and complementary WHO PV indicators to examine structure, process, outcomes, and identify the areas of strength and weakness of PV systems at different levels of development in three Arab countries (Jordan, Oman, and Kuwait). While previous studies have set out to provide an overview of the status and performance of Arab countries’ PV systems, this study goes beyond these studies to provide a deeper exploration of the study countries’ PV facilities, set-up dynamics, and outcomes. The use of a mixed-methods approach involving participants with intimate knowledge of PV policy and practice from both the NPVC and the PI in their respective countries permitted the identification of the implemented PV systems’ areas of best practice and challenges both qualitatively and quantitatively. The insights gained can be used for the development of a strategy towards improving patient safety through the development of a high performing PV system particularly within countries with systems at a nascent stage of development (such as Kuwait).

This study’s findings suggest that despite the presence of an operational PV system in all three countries, their performance and achievements require suitable and sustained improvement as they fall short in several indicators. The main factors impacting PV system performance in the three study countries fell into four main categories, namely, organisation and infrastructure; policy and resources; ADR reporting rates and signal detection; and stakeholder knowledge, awareness, and attitudes towards PV. In what follows, each of these key findings are discussed in detail in the context of the WHO PV indicators [15] and existing research concerning PV.

### 4.1. Organisation and Infrastructure

An important strength of Jordan and Oman’s PV systems was the presence of an officially recognised PV centre within the NMRAs’ organisational structure, thus providing them with increased visibility and significance. Contrastingly, the absence of an officially recognised PV centre weakened Kuwait’s PV system and prevented it from being operationalised. The existence of an officially recognised national PV centre provides visibility, acts as a reference point for stakeholder interaction, and provides an indication of the country’s commitment to accomplishing PV objectives [15]. National governments’ legitimisation of the NPVC acts as a facilitator to the mobilisation of the adequate and sustainable resources required for stable operation of the system [19].

In Jordan, the presence of a national PV advisory committee was seen as a strength, whereas its absence in Oman and Kuwait was considered a shortcoming. This meant that their NPVCs did not have the advantage of receiving expert feedback regarding drug safety issues. The WHO views such a committee as an integral part of the PV system [15] as its absence negatively impacts both system processes, e.g., risk assessment and management, as well as outcomes, e.g., regulatory actions [20].

The NPVCs in Jordan and Oman (but not Kuwait) both possessed access to a computerised case-report management system (VigiFlow), which offered the advantages of ensuring report accuracy and use of statistics. Access to VigiFlow allows for a cost-effective means of possessing a comprehensive (otherwise expensive) database with the added benefit of access to the WHO’s global ADR reporting data [21]. However, the finding that a high percentage of reports received were of low quality, combined with possibly limited NPVC staff’s expertise, meant that the data analysis option was not used, further emphasising the importance of targeted training for reporters and NPVC staff.

The WHO’s designation of the existence of a dedicated computer for PV activities and a computerised case-report management system as “complementary” indicators highlighted that the guidance may not adequately reflect the importance of technology in facilitating reporting and subsequent data management. Considering the advancement of, and access to, information technology globally, it may be time that these indicators be reclassified as “core”.

### 4.2. Policy and Resources

The government-enacted PV legislation in Jordan represented an important component of the country’s PV system, which granted it the authority to enforce and monitor implementation. Contrastingly, its absence in Kuwait and Oman was viewed as a weakness that deprived the centres of authority. The development of a national PV policy and other legislative instruments is an important measure to ensure the sustainability and effectiveness of PV structures [20]. Moreover, the presence of a clear legal framework accompanied by matching regulations ensures greater compliance and enforcement compared to relying on guidelines and normative practices which are not specifically binding [15,20].

A surprising finding was that despite all three countries not having a dedicated budget for PV, only in Jordan was this brought up as a shortcoming that deprived them of the ability to promote understanding, education, and training in PV or the hiring of additional staff. A possible explanation for this could be that both Kuwait and Oman are considered high-income countries, whereas Jordan is an upper-middle-income country according to the World Bank [22]. It is only when the other structural components of a PV system are paired with a regular and sustainable budget that real action and long-term planning can be achieved [23,24,25].

All three countries’ PV systems had human resources to carry out their functions. However, the number of employees assigned to carry out essential PV activities such as entering ADR reports into the national database, review of PV reports (i.e., PSURs and RMPs), and the conducting of training workshops and awareness campaigns was deemed insufficient by the NPVCs’ staff. The optimum staff number for a functional PV centre should be balanced against need and funds and must take into account the total population, scope of products, and the mode of PV activities [26]. Guidance from the WHO recommends that at least one of each of the following should be employed to support the full-time staff in carrying out the day-to-day PV activities: secretarial and data entry staff as well as an IT expert [27]. None of the studied countries were equipped with such personnel, thus placing an increased burden on existing staff.

### 4.3. ADR Reporting Rates and Signal Detection

Low ADR reporting rates represented a challenge for the three countries’ PV systems, resulting in local regulatory actions relying on decisions made in other countries. This suggests the information collected by the system is insufficient and/or inadequate to identify signals of drug-related problems and to support local regulatory decisions [28]. Under-reporting delays signal detection and, by extension, decision making to maintain an appropriate drug benefit/risk ratio [29]. The PV system’s ability to detect signals “underscores its relevance in identifying safety problems and promoting the safe use of medicines” [15] (p.33). On the other hand, the absence of regulatory actions points to a non-functional or dysfunctional system and a failure to monitor drug safety [15]. It might be beneficial to set up cross-country collaborative efforts with the goal of consolidated reporting to VigiBase [28].

### 4.4. Stakeholders’ Knowledge, Awareness, and Attitudes towards PV

In two out of the three (Jordan and Kuwait) countries’ national curricula for HCPs lacked PV, whereas in one (Oman), it was incorporated into the curriculum of only some HCPs (pharmacists and nurses), pointing to a deficiency in all three systems. This was believed to be a contributing factor to low awareness among HCPs. The absence of PV from the curriculum suggests HCPs’ lack of preparedness to deal with drug safety issues they will encounter during their practice [15]. Given HCPs’ responsibility to report ADRs during their practice, it is important that strategies that contribute to the promotion of PV by multidisciplinary teams in healthcare institutions be implemented [30]. Lack of undergraduate PV education and training contributes to low levels of knowledge, skills, and actions among HCPs [30,31,32]. These factors combined with negative attitudes have been linked to low and/or under-reporting of ADRs previously discussed here and confirmed by others [29,31,33]. Studies have demonstrated that the implementation of PV-related education/training as a module or course for HCPs increases their PV knowledge and improves their reporting practices [34,35,36,37]. Despite the WHO’s designation of PV as part of the curriculum as a “core” indicator, it may be advisable to designate this as a “complementary” indicator, and instead further emphasise a broader and longer-term strategy to ensure education in PV reporting, which would include HCPs’ curricula.

### 4.5. Study Limitation

This study has some limitations. Despite the WHO PV indicators’ usefulness as a tool for evaluating PV system performance, obtaining information on the indicators is dependent on facilities’ recordkeeping quality. Members of the NPVCs in the studied countries lacked awareness regarding measuring indices to monitor and evaluate PV system performance and therefore neither collected nor kept records of such data. Therefore, this limited the collection of information on some of the indicators. Assessment of some of the process and outcome indicators included as part of the tool require the assistance of individuals with expertise in areas such as diagnostics or health economics, which are not readily available in developing countries. The absence of the Jordanian PV system’s process and outcome indicators’ data prevented the study from presenting a more complete picture of the areas of its strength and weakness in comparison to the other two countries studied.

## 5. Conclusions

This study has shown that despite the recent progress made in the three Arab study countries’ PV systems, they still lacked several of the indices mentioned in the WHO’s guidance. Therefore, greater governmental prioritisation of PV as part of its public health policies’ portfolio through providing the necessary legislative enforcements, resources, and expertise as part of a well-structured system in each country is needed. Furthermore, more efforts are needed in coordinating regional efforts so that experience and expertise from advanced systems can be utilised in bolstering nascent systems. The Arab GVP guideline, with its aim of unifying PV procedures and activities among Arab countries [8], offers an opportunity to facilitate such efforts. The next steps that should be taken to improve PV systems in the Arab World include:Lobbying national governments and political parties on the importance of having a functional national PV system to obtain their commitment to supporting the system with legislation as well as suitable and sustained resources.Ensuring the establishment of key organisational and infrastructure elements including a dedicated and officially recognised NPVC, an expert advisory committee, and a computerised national database.Establishing training and development programmes addressing the key elements of PV and developing guidelines to support best PV practices among HCPs.

Future research should focus on evaluating the outcomes of the PV system, particularly hospitalisation, mortality, and financial impact as the current study demonstrated that these are areas for which information is particularly lacking.

## Figures and Tables

**Figure 1 ijerph-19-02518-f001:**
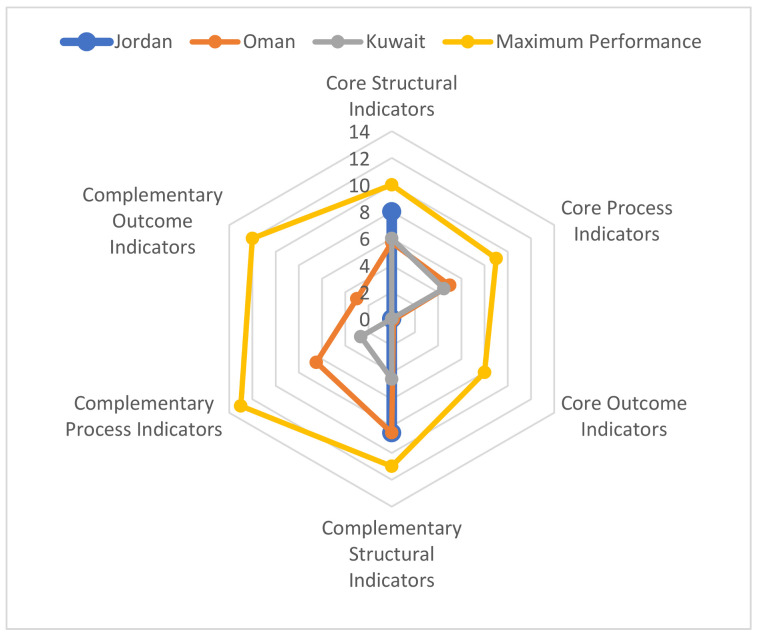
Six-axis radar diagram showing Jordan, Oman, and Kuwait’s pharmacovigilance systems’ scores for the six main categories of WHO pharmacovigilance indicators.

**Figure 2 ijerph-19-02518-f002:**
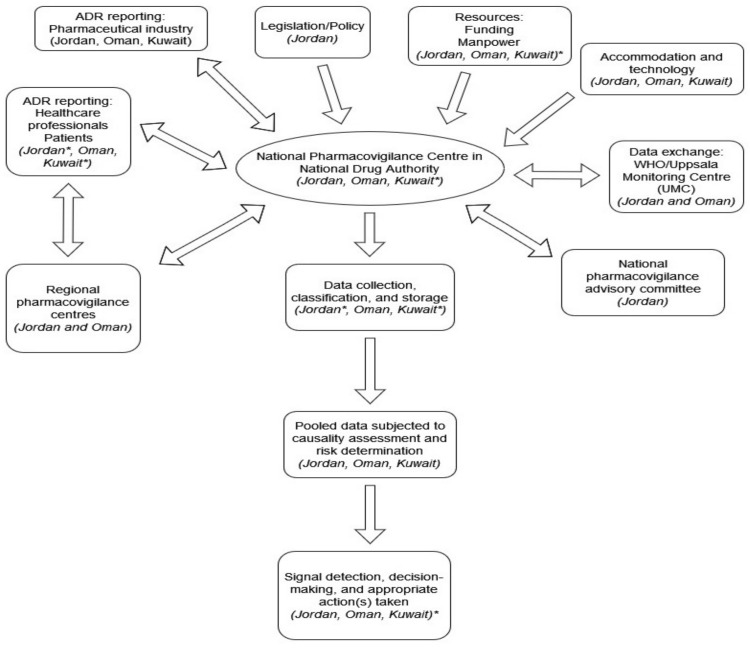
Areas of pharmacovigilance system strength and weakness in Jordan, Oman, and Kuwait. In the case where the pharmacovigilance system component is present, the county’s/countries’ name(s) is/are mentioned and vice versa in the case where the pharmacovigilance system component is absent. * Indicates pharmacovigilance system component present but represents an area of weakness based on study results.

**Table 1 ijerph-19-02518-t001:** Comparison of core structural WHO pharmacovigilance indicators’ performance in Jordan, Oman, and Kuwait.

Indicator Item	Assessment	Jordan	Oman	Kuwait
CST1	Existence of a pharmacovigilance centre, department, or unit with a standard accommodation	Rational Drug Use and Pharmacovigilance Department	Department of Pharmacovigilance and Drug Information	Quality Assurance Unit—not officially recognised
CST2	Existence of a statutory provision (national policy, legislation) for pharmacovigilance	Law titled “The Pharmacovigilance Directives”	Only “Guideline on GVP in Oman”	Only memos issued to companies
CST3	Existence of a medicines’ regulatory authority or agency	Jordan Food and Drug Administration (JFDA)	Directorate General of Pharmaceutical Affairs and Drug Control (DGPA&DC)	Kuwait Drug and Food Control Administration (KDFCA)
CST4	Existence of any regular financial provision (e.g., statutory budget) for the pharmacovigilance centre	No	No	No
CST5	The pharmacovigilance centre has human resources to carry out its functions properly	5 full-time employees	5 full-time employees	5 full-time and 1 part-time employee
CST6	Existence of a standard ADR reporting form in the setting	Yes	Yes	Yes
	CST6a—Availability of relevant fields in standard ADR reporting form to report medication errors	Yes	Yes	Yes
	CST6b—Availability of relevant fields in standard ADR reporting form to report suspected counterfeit/substandard medicines	Separate form	Yes	Separate form
	CST6c—Availability of relevant fields in standard ADR reporting form to report therapeutic ineffectiveness	Yes	Yes	Yes
	CST6d—Availability of relevant fields in standard ADR reporting form to report suspected misuse, abuse and/or dependence on medicines	Yes	Yes	Yes
	CST6e—Availability of a standard ADR reporting form for the general public	Same form as for HCPs	Same form as for HCPs	Same form as for HCPs
CST7	Existence of a process in place for collection, recording, and analysis of ADR reports	Yes	Yes	Yes
CST8	Incorporation of pharmacovigilance into the national curriculum of the various healthcare professions			
	CST8a—Medical doctors	No	No	No
	CST8b—Dentists	No	No	No
	CST8c—Pharmacists	No	Yes	No
	CST8d—Nurses or midwives	No	Yes	No
	CST8e—Others—to be specified	No	No	No
CST9	Existence of a newsletter, information bulletin, and/or website as a tool for dissemination of information on pharmacovigilance	Newsletter and website	No	Newsletter
CST10	Existence of a national ADR or pharmacovigilance advisory committee or an expert committee in the setting capable of providing advice on medicine safety	Health Hazard Evaluation Committee	No	No

**Table 2 ijerph-19-02518-t002:** Comparison of core process WHO pharmacovigilance indicators’ performance in Oman and Kuwait.

Indicator Item	Assessment	Oman	Kuwait
CP1	Total number of ADR reports in the previous year (2020)	1628	708
	CP1a—Total number of ADR reports received in the previous year (2020) per 100,000 people in the population	31.88 *	16.58 *
CP2	Current total number of reports in the national database	19,731	890 ^†^
CP3	Percentage of total annual reports acknowledged and/or issued feedback	N/A	100% (acknowledgement)
CP4	Percentage of total reports subjected to causality assessment in the previous year (2020)	N/A	58.9%
CP5	Percentage of total annual reports satisfactorily completed and submitted to the NPVC in the previous year (2020)	84.3%	58.9%
	CP5a—Of the reports satisfactorily completed and submitted to the NPVC, percentage of reports committed to the WHO database	84.3%	0
CP6	Percentage of reports of therapeutic ineffectiveness received in the previous year (2020)	0.80%	N/A
CP7	Percentage of reports on medication errors reported in the previous year (2020)	4.4%	N/A
CP8	Percentage of registered pharmaceutical companies that have a functional pharmacovigilance system	N/A	N/A
CP9	Number of active surveillance activities that are or were initiated, ongoing, or completed in the past 5 years	0	0

N/A indicates data not available; * calculated using World Bank country total population data for the year 2020; ^†^ figures on the basis of data entry from third quarter of 2019, with prior data lost.

**Table 3 ijerph-19-02518-t003:** Comparison of core outcome WHO pharmacovigilance indicators’ performance in Jordan, Oman, and Kuwait.

Indicator Item	Assessment	Oman	Kuwait
CO1	Number of signals detected in the past 5 years by the NPVC	0	0
CO2	Number of regulatory actions taken in the preceding year (2020) consequent to NPVC activities	2 *	N/A ^†^
	CO2a—Product label changes (variation)	-	N/A
	CO2b—Safety warnings on medicines	-	N/A
	CO2b(i)—To health professionals	-	N/A
	CO2b(ii)—To the general public	-	N/A
	CO2c—Drug withdrawals	-	N/A
	CO2d—Other restrictions on the use of medicines	-	N/A
CO3	Number of medicine-related hospital admissions per 1000 admissions	N/A	N/A
CO4	Number of medicine-related deaths per 1000 persons served by the hospital per year	N/A	N/A
CO5	Number of medicine-related deaths per 100,000 persons in the population	N/A	N/A
CO6	Average cost (USD) of treatment of medicine-related illness	N/A	N/A
CO7	Average duration (days) of medicine-related extension of hospital stay	N/A	N/A
CO8	Average cost (USD) of medicine-related hospitalisation	N/A	N/A

- Indicates data not provided; N/A indicates data not available; * on the basis of combination of local and external data; ^†^ indicated in interviews that actions had been taken on the basis of combination of local and external data.

**Table 4 ijerph-19-02518-t004:** Comparison of complementary structural WHO pharmacovigilance indicators’ performance in Jordan, Oman, and Kuwait.

Indicator Item	Assessment	Jordan	Oman	Kuwait
ST1	Existence of a dedicated computer for pharmacovigilance activities	Yes	Yes	Yes
ST2	Existence of a source of data on consumption and prescription of medicines	No	No	No
ST3	Existence of functioning and accessible communication facilities in the NPVC	Yes	Yes	Yes
ST4	Existence of a library or other reference source for drug safety information	Yes	Yes	No
ST5	Existence of a computerised case-report management system	VigiFlow	VigiFlow	No
ST6	Existence of a programme (including a laboratory) for monitoring the quality of pharmaceutical products	Yes	Yes	Yes
	ST6a—The programme (including a laboratory) for monitoring the quality of pharmaceutical products collaborates with the pharmacovigilance programme	Yes	Yes	Yes
ST7	Existence of an essential medicines list which is in use	Yes	Yes	No
ST8	Systematic consideration of pharmacovigilance data when developing the main standard treatment guidelines	Yes	Yes	No
ST9	The pharmacovigilance centre organises training courses for:			
	ST9a—HCPs	Yes	Yes	Yes
	ST9b—The general public	No	No	No
ST10	Availability of web-based pharmacovigilance training tools for:			
	ST10a—HCPs	No	No	No
	ST10b—The general public	No	No	No
ST11	Existence of requirements mandating MAHs to submit PSURs	Yes	Yes	Yes

**Table 5 ijerph-19-02518-t005:** Comparison of complementary process WHO pharmacovigilance indicators’ performance in Oman and Kuwait.

Indicator Item	Assessment	Oman	Kuwait
P1	Percentage of healthcare facilities with a functional pharmacovigilance unit (i.e., submitting ≥ 10 reports to the NPVC) in the previous year (2020)	70%	N/A
P2	Percentage of total reports sent in 2020 by the different stakeholders includes:		
	P2a—Medical doctors	8.9%	N/A
	P2b—Dentists	0	N/A
	P2c—Pharmacists	81.9%	N/A
	P2d—Nurses or midwives	0	N/A
	P2e—The general public	0.12%	N/A
	P2f—Manufacturers	8.8%	>95%
P3	Total number of reports received per million population per year (2020)	318.80 *	165.79 *
P4	Average number of reports per number of HCPs per year (2020) includes:		
	P4a—Medical doctors	198	N/A
	P4b—Dentists	0	N/A
	P4c—Pharmacists	1474	N/A
	P4d—Nurses or midwives	0	N/A
P5	Percentage of HCPs aware of and knowledgeable about ADRs per facility	N/A	N/A
P6	Percentage of patients leaving a health facility aware of ADRs in general	N/A	N/A
P7	Number of face-to-face training sessions in pharmacovigilance organised in the previous year (2020) for:		
	P7a—HCPs	2	0 ^†^
	P7b—The general public	0	0
P8	Number of individuals who received face-to-face training in pharmacovigilance in the previous year (2020):		
	P8a—Health professionals	55	0
	P8b—The general public	0	0
P9	Total number of national reports for a specific product per volume of sales of that product in the country (product specific) from the industry	N/A	N/A
P10	Number of registered products with a pharmacovigilance plan and/or a risk management strategy among the MAHs in the country	105	N/A
	P10a—Percentage of registered products with a pharmacovigilance plan and/or a risk management strategy from MAHs in the country	-	N/A
P11	Percentage of MAHs who submit periodic safety update reports to the regulatory authority as stipulated in the country	29%	14%
P12	Number of products voluntarily withdrawn by market authorisation holders because of safety concerns in 2020	6	7
	P12a—Number of summaries of product characteristics (SPCs) updated by market authorisation holders because of safety concerns	-	N/A
P13	Number of reports from each registered pharmaceutical company received by the NPVC in the previous year (2020)	N/A	N/A

- Indicates data not provided; N/A indicates data not available; * calculated using World Bank country total population data for the year 2020; ^†^ COVID-19 pandemic restricted carrying out face-to-face training sessions.

**Table 6 ijerph-19-02518-t006:** Comparison of complementary outcome WHO pharmacovigilance indicators’ performance in Oman and Kuwait.

Indicator Item	Assessment	Oman	Kuwait
O1	Percentage of preventable ADRs reported out of the total number of ADRs reported in the preceding year (2020)	3.54%	N/A
O2	Number of medicine-related congenital malformations per 100,000 births	1	N/A
O3	Number of medicines found to be possibly associated with congenital malformations in the past 5 years	2	N/A
O4	Percentage of medicines in the pharmaceutical market that are counterfeit/substandard	N/A	N/A
O5	Number of patients affected by a medication error in hospital per 1000 admissions in the previous year (2020)	N/A	N/A
O6	Average work or schooldays lost due to drug-related problems	N/A	N/A
O7	Cost savings (USD) attributed to pharmacovigilance activities	N/A	N/A
O8	Health budget impact (annual and over time) attributed to pharmacovigilance activity	N/A	N/A
O9	Average number of medicines per prescription	N/A	N/A
O10	Percentage of prescriptions with medicines exceeding manufacturer’s recommended dose	N/A	N/A
O11	Percentage of prescription forms prescribing medicines with potential for interaction	N/A	N/A
O12	Percentage of patients receiving information on the use of their medicines and on potential ADRs associated with those medicines	N/A	N/A

- Indicates data not provided; N/A indicates data not available.

## Data Availability

The data presented in this study are available in Supplementary File 3. Additional data are available on request from the corresponding author.

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
