# Peer review of "Strengths and Weaknesses of the Pharmacovigilance Systems in Three Arab Countries: A Mixed-Methods Study Using the WHO Pharmacovigilance Indicators"

_ijerph, 2022, doi:10.3390/ijerph19052518_

Round 1
Reviewer 1 Report
The authors present in this manuscript a study on strengths and weaknesses of the PV systems of three Arab countries, focusing on structures, processes and outcomes. The authors can be congratulated for their work, which is interesting and well exposed. The organization of the article is clear and the subject fits quite well to the scope of the journal. Bibliographical references are suitable and in sufficient number to illustrate this work properly.
I have a few remarks (mainly typos) and suggestions to bring to the attention of the authors:
- Please add a space before each bracket citing a reference to fit the typography requirements of the journal.
- Line 76: please write 17,820 km² with the ² in superscript.
- I believe that this publication would be strengthened if the main documents/tools that were used to collect the data, such as the self-administered questionnaire mentioned at line 107, were available as Supplementary Materials.
- Although tables are useful to sum up the collected information, synoptic schemes (for example representing the general organization of the PV system and showing the critical points considered as weaknesses) would be of great value to better illustrate the information given.
Reviewer 2 Report
General comments :
The aim of the study is to examine the structures, processes, and outcomes of the PV systems in three Arab countries using the WHO PV indicators with the objective to inform recommendations for Arab and other developing countries with nascent PV systems. But all three countries lacked a dedicated PV budget, so they do not really apply all the WHO recommandations. So, perhaps it would be more effective to first try to improve the official organisation in these countries. I do not really see how the results of the study, even if it is interesting, can improve the system. Perhaps the authors could more develop their objective in the introduction. And, in the conclusion, it could be interesting to propose next steps to improve the PV system.
The impact of educational intervention and of training of the healhcare professionals are not discussed. It could be interesting because it can improves the practices in PV.
Further comments :
- The questionnaire collecting quantitative measures was only completed by Oman and Kuwait's NPVCs. Why ?
- Table 2 indicated : « Comparison of core process WHO pharmacovigilance indicators' performance in Jordan, 275 Oman, and Kuwait. » But no results are preented for Jordan. Why ?
It is the same for table 3 and the core outcome indicators.
And, it is difficult to compare because many data lack, particularly in table 3.
- Table 4 is not commented
- Again, in table 5, the title indicates a comparison for the 3 countries but only 2 are presented, and many data are not provided, which makes it difficult to compare
It is the same for table 6
Yet, only Jordan’s system possessed legislation for PV. Jordan has an officially recognised PV center within the NMRAs' organisational structure thus providing increased visibility and significance. So, the authors could explain why no data are presented in the different tables for Jordan.
- Is the drug utilisation pattern different according to the country ? Because it could influence the number of adverse effect reported
Round 2
Reviewer 2 Report
The authors took into consideration the remarks of reviewers and improved the paper .